# Isolation and Establishment of a Highly Proliferative, Cancer Stem Cell-Like, and Naturally Immortalized Triple-Negative Breast Cancer Cell Line, KAIMRC2

**DOI:** 10.3390/cells10061303

**Published:** 2021-05-24

**Authors:** Rizwan Ali, Hajar Al Zahrani, Tlili Barhoumi, Alshaimaa Alhallaj, Abdullah Mashhour, Musaad A. Alshammari, Yasser A. Alshawakir, Omar Baz, Abdullah H. Alanazi, Abdul Latif Khan, Hassan Al Nikhli, Mohammed A. Al Balwi, Lolwah Al Riyees, Mohamed Boudjelal

**Affiliations:** 1King Abdullah International Medical Research Center (KAIMRC), Medical Research Core Facility and Platforms (MRCFP), King Saud bin Abdulaziz University for Health Sciences (KSAU-HS), Ministry of National Guard Health Affairs (MNGHA), Riyadh 11481, Saudi Arabia; alzahraniha6@NGHA.MED.SA (H.A.Z.); barhoumitl@ngha.med.sa (T.B.); alhallajal@NGHA.MED.SA (A.A.); mashhourab@NGHA.MED.SA (A.M.); bazom@NGHA.MED.SA (O.B.); 2Department of Pharmacology and Toxicology, College of Pharmacy, King Saud University, Riyadh 11451, Saudi Arabia; malshammari@ksu.edu.sa; 3Experimental Surgery and Animal Lab, College of Medicine, King Saud University, Riyadh 11451, Saudi Arabia; yalshawakir@ksu.edu.sa; 4Department of Pathology and Laboratory Medicine, King Abdulaziz Medical City (KAMC), MNGHA, Riyadh 11426, Saudi Arabia; AnaziA1@NGHA.MED.SA (A.H.A.); khanab4@ngha.med.sa (A.L.K.); nikhlih@ngha.med.sa (H.A.N.); BalwiM@NGHA.MED.SA (M.A.A.B.); 5KAIMRC, Department of Medical Genomic Research, KSAU-HS, Riyadh 11481, Saudi Arabia; 6College of Medicine, KSAU-HS, Riyadh 11481, Saudi Arabia; 7Department of Surgery, KAMC, MNGHA, Riyadh 11426, Saudi Arabia; lu_md@yahoo.com

**Keywords:** breast cancer, cell line, KAIMRC2, triple negative, stem cells, metastatic, characterization, drug treatment

## Abstract

In vitro studies of a disease are key to any in vivo investigation in understanding the disease and developing new therapy regimens. Immortalized cancer cell lines are the best and easiest model for studying cancer in vitro. Here, we report the establishment of a naturally immortalized highly tumorigenic and triple-negative breast cancer cell line, KAIMRC2. This cell line is derived from a Saudi Arabian female breast cancer patient with invasive ductal carcinoma. Immunocytochemistry showed a significant ratio of the KAIMRC2 cells’ expressing key breast epithelial and cancer stem cells (CSCs) markers, including CD47, CD133, CD49f, CD44, and ALDH-1A1. Gene and protein expression analysis showed overexpression of ABC transporter and AKT-PI3Kinase as well as JAK/STAT signaling pathways. In contrast, the absence of the tumor suppressor genes p53 and p73 may explain their high proliferative index. The mice model also confirmed the tumorigenic potential of the KAIMRC2 cell line, and drug tolerance studies revealed few very potent candidates. Our results confirmed an aggressive phenotype with metastatic potential and cancer stem cell-like characteristics of the KAIMR2 cell line. Furthermore, we have also presented potent small molecule inhibitors, especially Ryuvidine, that can be further developed, alone or in synergy with other potent inhibitors, to target multiple cancer-related pathways.

## 1. Introduction

Around the world, breast cancer is one of the leading causes of death in women. In the US, after skin cancer, breast cancer is the second most common cancer. Age, lifestyle, usage of oral contraceptives, and family history are the contributing factors in developing breast cancer [1,2,3]. Breast cancer-related mortality among Saudi women is high, with a prevalence index of 21.8%. It is also the second most common malignancy, accounting for the ninth leading cause of death in women. According to a recently accessed Saudi health council report, breast cancer accounted for 29% of all cancer types diagnosed in Saudi women [4,5].

Tumor relapse presents a significant challenge for effective treatment and contributes to increased mortality [6]. Studying the disease in vitro is a crucial step towards any in vivo investigation in understanding the disease and developing new therapy regimens. To date, the best and the most accessible model for studying cancer in vitro is to use immortalized cancer cell lines that possess oncogenic mutations and do not obey the Hayflick limit. Although human-derived organoids are gaining popularity, despite their limitations, cancer cell lines are still widely utilized in in vitro investigations of almost all cancer types, including breast cancer. These cell lines provide a rich source of knowledge for research related to cancer aggressiveness, genetics, progression, and ways to detect and treat the disease. Although there are many breast cancer cell lines available worldwide with their own unique mutated genetic makeup and specific characteristics, new cancer cell lines are always needed to understand the underlying heterogeneity of cancer.

Moreover, most of the available cancer cell lines are of Caucasian origin, and there are very few cell lines of Asian lineage, especially Arab origin. Therefore, a sizeable bank of cancer cell lines of Asian descent is much needed to study in-depth the diversity of this disease. A recent genomewide association study (GWAS) has accentuated the importance of diversity of ancestral genetic origin of cancer cell lines and showed that the available cancer cell lines database does not represent the diversity of cancer in the context of the distinct human population. For instance, it was observed that there are different frequencies of Breast Cancer type 1 and 2 (*BRCA1* and *BRCA2*) variants in Hispano/Latino and black populations. Another example is the higher frequency of Triple Negative Breast Cancers (TNBCs) among the women of West African ancestry [7].

It has previously been suggested that a small population of cancer cells express stem cell markers, including *CD133*, a membrane glycoprotein that has been used for isolating stemlike cells, and *ABCG2*, ATP binding cassette transporter [8]. These cancer stem cells (CSCs) are thought to be capable of self-renewal and are tumorigenic. It is also evident that CSCs are resistant to anticancer drugs and irradiation [9]. Thus, cancer cell lines that include CSCs have become a very attractive model for drug discovery and therapy development [10]. There are around 40 CSCs surface markers that have been identified and are providing molecularly targeted therapies for several cancers [11]. 

Not so long ago, we isolated and established a naturally immortalized breast cancer cell line, KAIMRC1 [12]. This study presents the establishment of another highly proliferative and spontaneously immortalized breast cancer cell line, KAIMRC2, with stem cell-like characteristics. This cell line has also been isolated from a Saudi breast cancer patient suffering from invasive ductal carcinoma. We envisage that this cell line, with its unique characteristics, will provide a powerful platform for future breast cancer research. Furthermore, pharmacological screening using these patient-derived cells, especially from different ethnic backgrounds, eventually helps us to understand the underlying genetic and functional mechanisms of the disease [13]. 

## 2. Materials and Methods

### 2.1. Clinical History

A 34-year-old female was diagnosed with a case of right breast invasive ductal carcinoma. Initially, the patient received neoadjuvant chemotherapy, four cycles of Epirubicin and Cyclophosphamide (EC) followed by three cycles of Docetaxel, Herceptin, and Pertuzumab with complete response. Right simple mastectomy resulted in SBR grade of 3, ER^−^/PR^−^ and BRCA^−^, HER-2^+^, and Ki-67 (MIB1) staining with a proliferation index of approximately 50%. It followed cycles of adjuvant radiotherapy and 17 cycles of adjuvant Herceptin. Later, she developed liver and bone metastases with liver function deterioration and elevation of liver enzyme levels. Afterward, the patient was maintained on Herceptin and Pertuzumab.

### 2.2. Collection and Culture of Breast Tumor-Derived Cells

Following the relevant guidelines set by the KAIMRC Ethics Committee for collecting and manipulating human samples, specimens were immediately obtained from primary tumor tissues after the surgical procedure, washed with Phosphate Buffered Saline (PBS), and prepared for pathological analysis. Informed written consent was obtained from the patient in accordance with the Institutional Review Board (IRB) of King Abdullah International Medical Research Center (KAIMRC), Riyadh, Saudi Arabia. Tissue samples were washed, minced into approximately 1–2 mm^3^ pieces, and incubated in 24-well cell culture plates at 37 °C and 5% CO_2_ atmosphere (Figure 1).

### 2.3. Cell Lines Culture

Human breast cancer epithelial cell lines, MDA MB-231 (HTB26) and MCF-7 (HTB-22), were purchased from ATCC, USA. KAIMRC1 and KAIMRC2 cell lines were isolated and established in the KAIMRC core lab facility. All the cell lines were maintained in advanced Dulbecco’s Modified Eagle Medium (DMEM) supplemented with 10% Fetal Bovine Serum (FBS), 50 units/mL penicillin and 50 μg/mL streptomycin, and 2 mM L-glutamine at 37 °C in a humidified 5% CO_2_ atmosphere. All were purchased from Gibco, New York, NY, USA.

### 2.4. Mycoplasma Assay

Mycoplasma contamination was assessed using the MycoAlert™ mycoplasma detection kit (Cat. #: LT07-118; Lonza, Basel, Switzerland). KAIMRC2 cells were cultured in the absence of mycoplasma active antibiotics for three days to maximize test sensitivity. Cell culture media supernatant sample was collected when the cells were 90–100% confluent. Mycoplasma detection was performed using the manufacturers’ instructions. Briefly, MycoAlert™ reagent was added to the sample (supernatant) in equal volume and incubated for 5 min, followed by measuring the luminescence by Molecular Devices SpectraMax plus 384 spectrophotometer (San Jose, CA, USA) (Read A). An equal volume of MycoAlert™ substrate was added to the sample and incubated for 10 min, and measured again (Read B). The ratio was less than 1, which is considered negative for mycoplasma.

### 2.5. Scanning Electron Microscopy (SEM)

Before SEM analysis, cells were fixed in 4% paraformaldehyde, graded concentrations of ethanol (Sigma, St. Louis, MO, USA) was used to dehydrate the samples before transfer to carbon taped stubs (Ted Pella, Redding, CA, USA). To enhance the electron conductivity, the samples were sputter coated with gold/palladium (Au/Pd) using Quorum Q300T D equipment (Laughton, East Sussex, UK). The images were acquired on an FEI NanoSEM 450 scanning electron microscope (FEI Company, Hillsboro, OR, USA) at 10 kV (Figure 1b,c). 

### 2.6. Transmission Electron Microscopy (TEM)

Cells were fixed in 4% glutaraldehyde (Cat # G5882, Sigma-Aldrich, Darmstadt, Germany) followed by fixation in 1% Osmium tetraoxide (OsO_4_) (Cat# 75632, Sigma-Aldrich) for 1 h each. The specimen was then dehydrated in the graded concentration of acetone and infiltrated with acetone: resin at 1:1 for 1 h, and then at 1:2 for 3 h. The sample was embedded in epoxy resin (Araldite) and polymerized overnight at 80 °C. Ultrathin sections of the specimen were obtained with Ultramicrotome (RMC Boeckeler, Tucson, USA), mounted on copper grids, and stained for contrast with uranyl acetate and lead citrate. Images were collected on a JEOL-JEM 1200 (JEOL Ltd., Peabody, MA, USA) operating at 120 kV with Digital Micrograph Imaging software (Figure 1d,e).

### 2.7. Cell Growth Curve

KAIMRC2 cells were seeded at a density of 5000 cells/well in the media described above into 24 well plates and incubated at 5% CO_2_ for 15 days. The media was replaced every 3rd day. Cells were collected and counted every 24 h in triplicates using Vi-Cell XR (Beckman Coulter, Brea, CA, USA). Cells reached a growth plateau by the 12th day, and cell death was apparent afterward. The cell growth curve was generated in MS Excel (Microsoft Office Profession Plus 2013) (Figure 2).

### 2.8. Wound Healing Scratch Assay

This assay was performed to determine the migration potential of the KAIMRC2 cell line. Cells were seeded at a high density in a 35 mm µ-dish (Ibidi, Planegg, Germany) and incubated at 37 °C for 1–2 days to obtain a confluent cell layer. A scratch was introduced in the cell layer, and the cell migration was observed for 24 h using timelapse imaging under the EVOS FL Auto microscopy system (Thermo Fisher Scientific Molecular Devices, Waltham, MA, USA). Images were acquired every hour and wound healing potential was analyzed using Image J software (NIH, New York, NY, USA) (Figure 2a–g).

### 2.9. Immunocytochemistry (ICC)

Cells were seeded 24 h before the experiment in a 6-well dish as described above in the cell culture section. Cells were fixed, in some cases, permeabilized and incubated overnight at 4 °C with all the antibodies mentioned above in the flow cytometry paragraph. Counterstaining was performed with HOECHST 33,342 (Life Technologies, Life Technologies, Carlsbad, CA, USA) to detect the nucleus. All the staining was performed in triplicate and repeated twice. The plate was imaged using the EVOS FL Auto Microscope System (Figure 2p). 

### 2.10. Colony Formation Assay

Colony formation potential was examined by plating 100 cells on 6-well plates (Corning). Colony formation was monitored microscopically, and transmitted light images were acquired thrice a week. Colonies were stained with Hematoxylin and Eosin on Day 10, and images were acquired using Molecular Devices EVOS FL Auto system (Thermo Fisher Sceintific, Waltham, MA, USA) (Figure 2h–m).

### 2.11. Western Blot Analysis

MCF-7, MDA MB-231, KAIMRC1, and KAIMRC2 cells were seeded in 6-well plates in complete advanced DMEM for 24h. Before protein extraction, cells were pre-incubated with 10% serum-containing complete advanced DMEM and serum-free DMEM conditions for another 24 h. The Western blotting analysis was performed with mouse monoclonal antibody against mTOR (215Q18; Cat # AHO1232, Thermo Fisher Scientific; 1:500), Rabbit polyclonal antibody to Phopho-mTOR (Ser2448; Cat # 44-1125G, Thermo Fisher Scientific; 1:500), Mouse monoclonal antibody to Akt pan (40D4; Cat # 2920s, Cell Signaling, Danvers, MA, USA; 1:2000), Rabbit monoclonal antibody to Phopho-Akt (Ser473; D9E XP^®^; Cat # 4060s, Cell Signaling; 1:1000), Rabbit monoclonal antibody to Integrin β4 (Cat # 4707s, Cell Signaling; 1:1000), Rabbit monoclonal antibody to Cyclin-D1 (92G2; Cat # 2978; Cell Signaling; 1:1000), Rabbit polyclonal antibody to p38 MAPK (Cat # 9212s; Cell Signaling; 1:1000), Rabbit monoclonal antibody to Oct4A (C52G3; Cat # 2890s; Cell Signaling; 1:1000), Rabbit monoclonal antibody to STAT5 (3H7; Cat # 4459s; Cell Signaling; 1:1000), Rabbit monoclonal antibody to JAK2 (18H11L8; Cat # 702434; Invitrogen; 1:1000), Rabbit monoclonal antibody to E-Cadherin (24E10; Cat # 3195s, Cell Signaling; 1:1000), and, sample loading was examined by probing with mouse monoclonal antibody against β-Actin loading control (BA3R; Cat # MA5-15739, Thermo Fisher Scientific; 1:500). Signals were detected using a ChemiDoc MP System (Bio-Rad, Hercules, CA, USA) and analyzed on ImageLab and MS Excel 2013 software (Figure 3a).

### 2.12. Human Proteome Profiler Arrays

To further profile the proteins involved in KAIMRC2 proliferation and immortalization, several Human Proteome Profiler™ (R&D Systems) antibody arrays were utilized. Human Phospho-Kinase Array (Catalog # ARY003B), Human Pluripotent Stem Cell Array (Catalog # ARY010), Human Phospho-RTK Array (Catalog # ARY001B), Human Phospho-Immunoreceptor Array (Catalog # ARY004B), and Human Cytokine Array (Catalog # ARY005B). All were purchased from R&D Systems, Minneapolis, MN, USA, and used according to the manufacturer’s protocol. KAIMRC2 cells were seeded in 6-well plates in complete advanced DMEM for 24h. Before the experiment, cells were pre-incubated with 10% serum-containing complete advanced DMEM and serum-free DMEM conditions for another 24 h. Cells were lysed, and the lysate was processed for all the profilers except Human Cytokine Array, where supernatant was collected and processed for the proteome profiler. The membranes were imaged using ImageLab software, a ChemiDoc MP System (Bio-Rad), and the data were quantified by Gel Analyzer, Image J Software v 1.53 (NIH, New York, NY, USA) (Figure 3b). 

### 2.13. Sample Preparation and Analysis of Real-Time qPCR

Total RNA was isolated from KAIMRC1, KAIMRC2, MDA MB-231, and MCF-7 cells using the RNA Isolation Kit (Ambion) following manufacturers’ instructions. RNA was quantified on Nanodrop 8000 Spectrophotometer (Thermo Scientific, Waltham, MA, USA). Briefly, Applied Biosystems’ high capacity cDNA reverse transcription kit was used to reverse transcribe1µg of total RNA into cDNA. The resulting cDNA was diluted 1:2 in ddH_2_O. Commercially available Human Breast Cancer RT^2^ Profiler PCR Array (Applied Biosystems^®^, Foster City, CA, USA) was employed. The samples were processed following the manufacturer’s protocol. PCR amplification was performed on a 7900HT Fast Real-time PCR System (Applied Biosystems^®^, Foster City, CA, USA). Data were analysed on RQ Manager 1.2.1 software (Applied Biosystems^®^, Foster City, CA, USA). Microsoft Excel 2013 was used for further analysis and the ΔΔ_Ct_ method was utilized to calculate relative changes in mRNA expression levels (Figure 4).

### 2.14. MTT Cell Proliferation Assay

KAIMRC2 cells were seeded at 5000 cells/well into 96-well plates and incubated at 5% CO_2_ incubator overnight. Subsequently, cells were treated with compounds in serial dilution followed by incubation for 48 h. MTT assay was performed after 48 h, and the absorbance was read at 560 nm using Molecular Devices Spectra Max Plus 384 Spectrophotometer. Background readings were collected at 620 nm. Absorbance readings were normalized and expressed as a relative percentage. The data were analyzed in Graphpad Prism software, and the half-maximal inhibitory concentration (IC_50_) was determined (Figure 5a,b). Error bars denote standard deviation (SD). The statistical significance of differences was determined using the Student’s t-test. *p* < 0.05 was considered statistically significant.

### 2.15. Flow Cytometry Analysis

Immunophenotyping of the KAIMRC2 cell line was performed on the FACS Canto II flow cytometer (BD). Fluorescently labeled primary antibodies CD49f, CD49c, EpCAM (EBA-1), CD44, CD24, CD227, CD47, HER2, CD133, CD34, ALDH1A1 (LSBio), CD117, Vimentin (Thermo Scientific), and CD234 (E-Cadherin) were purchased from BD Pharmingen unless otherwise specified. Briefly, cells were pelleted and incubated with antibodies for 1 h and analyzed on the flow cytometer. Cellular results were sorted on histogram plots for positive cell populations in FITC and PE channels (Figure 6e). To isolate population of human breast cancer stem cells we utilized R&D Systems MagCellect™ Human CD24^−^CD44^+^ Breast Cancer Stem Cells Isolation Kit (Biotechne^®^, Minneapolis, MN, USA). This technique is a two-step procedure combining both negative CD24 and positive CD44 selection to isolate the cancer stem cells (Appendix A).

### 2.16. Karyotyping

Karyotype was produced according to standard procedure, as described in [14]. Briefly, the cells were harvested, centrifugated at 1000 rpm for 4 min, hypotonic 0.5% KCl solution was added for 30 min, followed by fixative 2× (methanol/acetic acid 3:l) at room temperature for 30 min. Cells were spread on slides, air-dried, stained with Giemsa stain for 20 min, followed by the analysis on a light microscope at 10× and 100× magnification (Figure 6f).

### 2.17. Histopathology

KAIMRC2 cells sample were fixed with 4% paraformaldehyde, placed in cassettes, and mounted into an automated vacuum tissue processor (Leica Biosystems, Nußloch, Germany). The samples were embedded and blocked-in paraffin wax, and thin sections were made using a microtome (Accu-Cut^®^ SRM™, Sakura, Alphen aan den Rijn, The Netherlands). Sections were stained with Mayer’s hematoxylin solution and counterstained in eosinphloxine solution. Slides were examined and imaged under an upright microscope (Eclipse TS100, Nikon, Tokyo, Japan) (Figure 6c,d).

### 2.18. Mice Model

Animal studies were approved by the KAIMRC Animal Ethical Committee, Institutional Animal Care and Use Committee (IACUC, Riyadh, Saudi Arabia), and the National Committee of Bio-Ethics (NCBE, Riyadh, Saudi Arabia). All the procedures during the animal studies were performed following the guidelines set by the Research Ethics Committee (REC, Riyadh, Saudi Arabia) of King Saud University (KSU, Riyadh, Saudi Arabia). Xenograft studies were conducted in four- to six-week-old female non-obese diabetic/severe combined immune-deficient (NOD/SCID) mice (NCI-Frederick, Frederick, MD, USA). A total of 10 mice were used for the study. The cells (1,000,000, 100,000, 1000, 500 or 100 in 50 µL of ice-cold HBSS: Matrigel *v*/*v*) were injected through the 27-gauge needle using a Hamilton syringe. The animals’ tumor burden was assessed weekly by measuring the animal weight and body score, as described by [15]. Tumor-bearing mice were monitored three times a week until a palpable tumor nodule was felt and then daily after that. Tumor diameter was measured using a digital caliper, and the tumor volume was estimated using the formula (Volume = (Width) ^2^ * Length/2). Mice were sacrificed after three months. Mice were sacrificed as previously described [16], briefly anesthetized with 3% isoflurane mixed with O_2_ at 1 L/min. The depth of anesthesia was confirmed by the rear foot squeezing, then a cardiac puncture and diaphragm removal was performed under complete anesthesia. Tumor weight was weighed after surgery. Image segmentation, quantitative image analysis, and statistical analysis will be performed in Image J software (NIH, New York, NY, USA) (Figure 6a,b).

## 3. Results

### 3.1. Isolation and Establishment of KAIMRC2 Cell Line

A tissue sample after mastectomy was obtained from a 34-year Saudi Arabian female breast cancer patient with invasive ductal carcinoma. The tissue sample was minced into small slices, approximately 1 mm–2 mm, and cultured in advanced DMEM. Continuous culture of tissue samples resulted in fibroblastlike cells adhering to the culture-ware as well as outgrowth of cells attached to the tissue on the cell culture plate was observed. The outgrowth of large primary tumor fibroblastlike cells was visible within a few days (Figure 1f). These cells were passaged for more than 15 times until they transformed into smaller and cuboidal epithelial-like cells. The duplication time increased, and cell size also reduced to a few micrometers (Figure 1g–i). These cells were regularly passaged and expanded in the standard cell culture media for characterization. The newly developed cell line, KAIMRC2, was stably passaged and is currently in its 65th passage. These cells adhere firmly to the glass and plastic surfaces, unlike KAIMRC1 cells.

Interestingly, during initial attachment to any surface, lipid blebs form at the edge of the cell membrane. These blebs circle within the cell membrane around the cell (Figure 1 j–o). Further research is needed to understand the unique behavior of these cells during cell–surface interaction. A video is available in the Appendix A.

### 3.2. Growth Characteristics

The growth kinetics of the KAIMRC2 cell line at passages 30–35 were studied. The growth curve of the KAIMRC2 cell line is shown in Figure 2n. KAIMRC2 cells showed rapid growth with a doubling time of ~16 h, whereas the population-doubling times of other well-established breast cancer cell lines, MDA MB-231, MCF-7, MCF-10A, and KAIMRC1 cells were ~24, 45, 18, and 24 h, respectively [12].

### 3.3. Wound-Healing Assay

Scratch or wound healing assay was performed to assess the migration potential of KAIMRC2 cells as shown in Figure 2a–g and compared with previously published wound healing assay data of other breast cancer cell lines, including MCF-7, KAIMRC1, MDA MB- 231, and the normal MCF-10A [12]. Live-cell time-lapse microscopy was utilized to monitor cell migration and wound closure over time. The KAIMRC2 cells showed wound healing potential close to 8–10 h (Figure 2a), which was the fastest among other breast cancer cell lines in consideration.

### 3.4. Colony Formation Assay

The ability of cells to make colonies in an anchorage-independent manner was assessed. KAIMRC2 cells were seeded at a density of 100 cells per 6-well plates (Figure 2h–m). Single cells were spotted and marked for future reference. KAIMRC2 cells formed small colonies that were visible after two days of single-cell culture. In the next eight days, there was a significant increase in the number of large colonies was observed. Extra-large sized colonies with a necrotic center were visible after ten days of culture (Figure 2o).

### 3.5. Chromosome Analysis

Karyotyping of KAIMRC2 cell line uncovered a complex karyotype including; hexasomy of chromosome 9; tetrasomy of chromosomes 1, 2, 5, 6, 7, 11, 12, 17, 18, and 21; trisomy of chromosomes 3, 8, 14, 15, 16 and 19 (Figure 6f). These cells have complicated structural abnormalities, in addition to the numerical abnormalities that explain the aggressive pathogenicity of the disease.

### 3.6. Immunophenotyping

Flow cytometry (Figure 6e) and immunocytochemistry (Figure 2p) were utilized to screen a panel of cell surface and stem and cancer cell markers to study the KAIMRC2 cell line phenotype. As evident from Table 1, KAIMRC2 cells highly express CD47, CD116 (GM-CSFR), CD49c (ITGA3), CD324 (E-cadherin), Aldehyde Dehydrogenase 1 (ALDH1A1), and CD44 suggesting a breast epithelial phenotype. Moreover, these cells express the CSCs markers CD133, Stage-Specific Embryonic Antigen-1 (SSEA-1), CD49f (ITGA6), and CD326 (EpCAM), hinting towards the presence of a subpopulation of CSCs. To confirm the presence of CSCs we also utilized MagCellect™ Human CD24^−^CD44^+^ Breast Cancer Stem Cells Isolation Kit. The CD24^−^CD44^+^ cells were then analysed using flow cytometry (Appendix A). The comparative analysis of expression markers suggests that the KAIMRC2 cell line is sufficiently different from other breast adenocarcinoma cell lines. Elevated expression of CD194 (CCR4), CD116, α-SMA, and CD44 points toward the metastatic potential of these cells, and it correlates well with the patients’ data showing liver and bone metastasis.

Moreover, the expression of progesterone receptor (PR), estrogen receptor (ER), and human epidermal growth factor receptor 2 (HER2) was also analyzed by immunocytochemistry to classify KAIMRC2 cells. The cells were found to be triple-negative. This result was validated using Western blot (data not shown).

### 3.7. Molecular Characterization of KAIMRC2 Cell Line

We executed several cellular assays, including selected breast cancer gene profiling, phosphoprotein profiling and drug-induced sensitivity assays to investigate possible pathways involved and responsible for the natural transformation of the KAIMRC2 cell line.

### 3.8. Phospho-Protein Profiling

Immunoblotting analysis of key AKT pathways and other signaling proteins revealed that similar to KAIMRC1 cells, the KAIMRC2 cell line also showed constitutive, i.e., ligand-independent oncogenic signaling protein activation, AKT (Figure 3a). It also showed constitutive activation of mTOR, which implies that the AKT/mTOR pathway has a significant role in the cell lines’ natural immortalization. AKT is a crucial regulator of protein translation, transcription, cell proliferation, and apoptosis [17,18]. Phosphorylation of AKT is key to the activation of mTOR. 

An in-depth study of the phosphorylation of several key human proteins was performed. Various human protein profiler arrays, i.e., Phospho-Kinase Array, Pluripotent Stem Cell Array, Phospho-RTK Array, Phospho-Immunoreceptor Array, and, Cytokine Array were utilized (Figure 3b). The results demonstrated an upregulated expression of insulin receptor (IR), GATA-4, PDX-1, E-cadherin, and SERPINE1. Interestingly, Cyclin D1 is highly expressed in the starvation condition in KAIMRC2 cells. Figure 3c shows an illustration of our proposed model of the KAIMRC2 cell line survival pathway.

### 3.9. Drug Sensitivity

To assess the KAIMRC2 cell line’s drug-related sensitivity, we tested panels of commercially available kinase inhibitors, stem cell modulators, and epigenetic regulator compounds (Appendix A). In the initial phase, these compounds were screened at 10 µM concentration of each compound in duplicates for 48 h to identify candidate compounds. Cell viability was assessed using MTT assay, and the survival was measured in percentage relative to DMSO control. Figure 5a shows all those compounds that presented less than 60% cell viability. Among these, 16 compounds were selected to calculate their IC_50_ (Figure 5b) further. Table 2 shows the IC_50_ values of the selected compounds.

### 3.10. Gene Expression Analysis

Here, we have used a commercially available breast cancer gene panel to profile the expression of key genes involved in breast carcinogenesis. We found 51 genes showing significant gene expression changes in KAIMRC2 cells relative to normal breast MCF-10A cells (Figure 4a). It is noteworthy that the normal breast MCF-10A cells, used as reference cells, had a different ethnic background (Caucasian) than the KAIMRC2 cells. The key up-and downregulated genes were segregated. Reactome Pathway database (reactome.org (accessed on 15 January2020), a web-based pathway analysis tool, was used to discover the molecular pathways affected by differentially regulated genes. In KAIMRC2 cells, several upregulated genes were found to be activating Oncogenic MAPK signaling, cell cycle-related pathways, signaling by nuclear receptors, and transcriptional regulation by TP53 (Figure 4b), whereas downregulated genes were mainly involved in the signaling by Interleukins, transcription and cell cycle checkpoints (Figure 4c).

### 3.11. In Vivo Tumorigenicity and Stemness

We injected the KAIMRC2 cell line in at least five mice to confirm their tumorigenic (10^6^ to 10^7^ cells/mice) and stemness (500–1000 cells/mice) potential. Mice were sacrificed on the 26th day after injection. Figure 6a shows an image of nude mice after 18 days of injection. A noticeable tumor lump is visible in the picture, whereas the same mice’s appearance on the 26th day showed a big tumor lump. Table 3 contains a detailed account of tumor size, dimension, and weight of 3 representative mice. The tumor was isolated and stained with H&E. The histological section of isolated tumors exhibited all the malignancy hallmarks, i.e., nuclear pleomorphism, hyperchromasia, clumped nuclear chromatin, and mitosis. Extensive tumor coagulative necrosis and the concentration of viable cancer cells were also visible around the blood vessels, i.e., peritheliomatous necrosis, a high-grade tumor hallmark.

## 4. Discussion

The complexity and heterogeneity of cancer have perplexed the scientific community around the globe for decades. Deciphering the complex environment of a cancer cell is the ultimate dream of a cancer biologist. Cell lines provide a rich resource to characterize cancer based on its genome, proteome, and molecular footprints. It is still challenging to find the right cell line that can provide all the required answers. Moreover, highly passaged cell lines are known to alter their morphology, gene, and protein expression profiles. It is crucial to establish new cell lines from the primary tumors with low passage numbers and unique cancer-related characteristics. 

It is well established now that CSCs are present in many cancer cell lines and play a pivotal role in the initiation and proliferation of aggressive cancer cells. Similar to cancer cells, CSCs are also believed to be immortal. CSCs are like seeds around which cancer cells grow and metastasize. Therefore, this small subset of cancer cells needs to be targeted to increase anticancer therapies’ efficacy. Due to the limited size of a tumor sample, it is challenging to isolate relevant tumor cells that can give rise to a cell line that can contemplate the tumor environment’s heterogeneity. It is even more challenging to establish CSCs from relevant cancer cell populations. Therefore, cancer cell lines with a subpopulation of CSCs provide a unique opportunity to study cancer cells along with their precursor cells.

We have isolated and established an aggressive breast cancer cell line with unique CSCs-like characteristics in this work. Immortalization was confirmed with continuous passaging of the cell line, and molecular characterization was performed using immunofluorescence, karyotyping, protein profiling, drug tolerance, and gene expression studies. KAIMRC2 cell line showed a very complex and abrupt chromatogram. Aneuploidy across all the chromosomes and the X-chromosome, especially trisomy, was revealed, a hallmark of aggressive cancer type.

High expression of CD47, CD49c, CD44, and ALDH1A1 suggests the breast cancer origin of the KAIMRC2 cell line. Overexpression of CD44, CD49c, CD116, ALDH1A1, EpCAM, SSEA-1, and Oct4 was the evidence of the presence of a subpopulation of CSCs in the KAIMRC2 cell line. Briefly, it has been earlier demonstrated that the frequency of tumor-initiating cells is ten times higher in the EpCAM^+^ fraction of breast CSCs than in the EpCAM^−^ fraction. Increased levels of EpCAM can induce bone metastasis [19]. Elevated ALDH1A1 expression is associated with breast tumorigenesis and metastasis [20]. CD194 or C-C Chemokine receptor type 4 (CCR4) promotes breast cancer lung metastasis by downregulation of T regulatory cells [21,22], CD116 or Granulocyte-macrophage colony-stimulating factor receptor (GM-CSFR) has also been shown to promote bone metastasis [23], and α-smooth muscle actin (α-SMA) is required for metastatic potential of human lung adenocarcinoma [24]. CD44, a cell-surface glycoprotein involved in cell–cell interactions, cell adhesion, and migration, can interact with MMP9 to promote tumor growth and metastasis [25]. An increase in the expression of MMP9 has shown metastatic potential and excessive cell proliferation. MMP9 degrades collagen IV and Laminin 5 and, in turn, augments the escape of metastatic cells through the basement membrane [26]. Gene expression of MMP9 is elevated along with the expression of CD44 in KAIMRC2 cells, reflecting the metastatic potential of the cell line. Interestingly, the knockdown of MMP9 triggers the switching of CD44 to an isoform resulting in a less invasive phenotype [27].

Gene expression profiling is a valuable tool to classify tumors into their relevant type and look for patterns among the group of genes to understand cancer’s heterogeneity. Gene expression profiling of the KAIMRC2 cell line revealed significant upregulation of *ABCB1* transporter, *ADAM23, CDKN2A, JUN, KRT19*, and *TFF3*, whereas considerable downregulation of *IGFBP3*, *KRT5*, *MUC1*, *SERPINE1*, and *THBS1* genes. Upregulated genes are implicated in Oncogenic MAPK signaling and cell cycle-related pathways. On the other hand, downregulated genes are involved in interleukin signaling and gene transcription.

It has been reported that several cytokines, including GM-CSF, IL-6, IL-8, and macrophage migration inhibitory factor (MIF), are highly expressed by the cancer-associated MSCs [28]. MIF, IL-8, and GM-CSF are all highly expressed by KAIMRC2, confirming its stem cell-like phenotype. Another cytokine highly expressed in KAIMRC2 cells is plasminogen activator inhibitor-1 (PAI-1) or SERPINE1, a serine protease inhibitor. It is present in increased levels in various disease states, including cancer. Although our gene expression analysis data showed downregulation of the *SERPINE1* gene, it is well established that gene expression does not always correlate well with the protein expression for several reasons, for example, a lower rate of mRNA transcription compared to protein translation [29,30].

AKT, a serine-threonine protein kinase, is at the center of multiple signaling pathways regulating cell growth, differentiation, and survival. There exists a reciprocal relationship between the phosphorylation of AKT and mTOR signaling. Insulin/insulinlike growth factor signaling (IIS) is the crucial pathway that responds to nutrient signals. The deregulation of this pathway can lead to type 2 diabetes, obesity, and cancer [31,32]. IIS mediated phosphorylation of AKT-T308 activates mTORC1 and mTORC2 phosphorylates AKT on S473. In brief, the IIS pathway is activated by the insulin [33] or insulinlike growth factor (IGF) binding to the insulin/IGF receptor. Upon stimulation by *IGF*, the tyrosine residues of the cytosolic domains are phosphorylated, which leads to the activation of the PI3K/Akt/mTOR pathway. The other pathway that complements and contributes to AKT/mTOR is the JAK/STAT pathway.

Interestingly, both of the pathways seem to be hyperactive in the KAIMRC2 cell line. For instance, the highly expressed cytokine, GM-CSF, might activate the JAK/STAT pathway, consequently contributing to cell survival. On the other hand, phosphorylation of IR/IGF-R might be contributing to the activation of the AKT/mTOR pathway, resulting in increased cell proliferation. Here, we propose that the constitutive activation of the AKT/mTOR pathway and the expression of the JAK/STAT pathway might be the reason for the aggressive proliferation and metastatic potential of the KAIMRC2 cell line. 

In our immunoblotting results, the KAIRMC2 cell line has exhibited Cyclin D1 expression in the starvation condition. Both JAK/STAT and AKT/mTOR pathways are involved in the activation of Cyclin D1, which is responsible for cell growth and proliferation. Another exciting observation while comparing KAIMRC1 and KAIMRC2 cell lines was the remarkable similarity between the protein expression profiles of these two cell lines (Figure 3a and Appendix A). These results suggest that there are marked differences among the cell lines originating from different ethnic backgrounds. New models based on other population groups are needed to devise a strong therapy regimen. 

This work has also shown that these cells produce hypertrophic lipid droplets/vesicles during their initial attachment period to a substrate. We are still unable to explain this phenomenon. It might be possible that these cells have the inherent characteristic of adipocyte cells because the primary tumor section was isolated from an adipose tissue area. Further research is needed to understand this process.

The ultimate goal of isolating and establishing a cell line is to understand the mechanism of cancer and to test the efficacy of a drug quickly and reliably. Keeping this in mind, we have tested a range of commercially available and in-house developed compounds (data not shown) to assess the drug-related sensitivity of the KAIMRC2 cell line. A list of the most potent inhibitors of KAIMRC2 cell growth is available in Table 2. These compounds are either kinase inhibitors or epigenetic regulators. Epigenetic regulation that involves DNA methylation and histone modifications play an essential role in gene transcription and, eventually, gene expression in eukaryotic cells. KDM5 family of enzymes function as demethylases and are strongly linked with oncogenesis [34]. It has been shown that there is a strong association between KDM5A expression in breast cancer and drug tolerance. Ryuvidine, a derivative of benzothiazoledione, has been recently identified as an inhibitor of KDM5A [35].

Interestingly, our results showed that Ryuvidine strongly inhibits KAIMRC2 cell growth at an IC_50_ of 0.8 µM. To our knowledge, this is the first report of the effect of Ryuvidine on a breast cancer cell line. Ryuvidine might be simultaneously inhibiting KDM5A and the cyclin-dependent kinases (CDK) pathway to arrest cell growth in the KAIMRC2 cell line. Further research is needed to validate this small molecule inhibitor’s mechanism in breast cancer cell lines. Other potent KAIMRC2 cell growth-inhibiting candidates were histone modifying enzyme inhibitors such as M344, a histone deacetylase inhibitor, and JIB 04, a histone demethylase inhibitor.

In conclusion, our results confirmed an aggressive phenotype, metastatic potential, and cancer stem cell-like characteristics of the KAIMR2 cell line. Furthermore, we have also presented potent small molecule inhibitors, especially Ryuvidine, that can be further developed alone or synergy with other potent inhibitors to target multiple cancer-related pathways.

## Figures and Tables

**Figure 1 cells-10-01303-f001:**
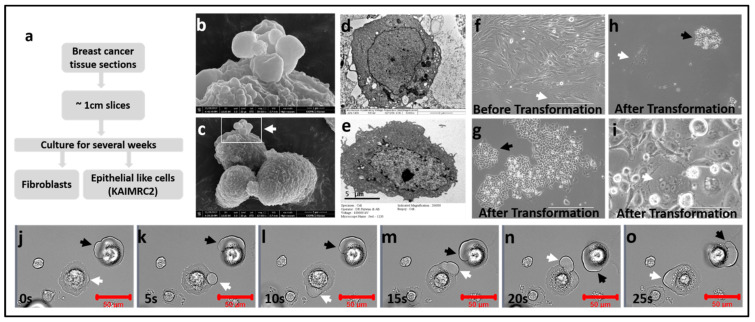
Microscopic characterization of the KAIMRC2 cell line. (**a**) Schematic representation of the workflow of isolation and culture of cancerous cells. Briefly, collected tissue sections were minced into approximately 1 cm pieces, followed by several weeks of growth in normal cell culture media. Large fibroblastlike cells were passaged for almost 15 times. Afterward, smaller sized, cuboidal, and epithelial-like KAIMRC2 cells started to make colonies. (**c**) SEM image of the KAIMRC2 cell line with a large-sized lipid droplet formation on its surface. Scale bar = 5 µm. (**b**) Enlarged image of lipid droplet on the surface of the cell membrane. Scale bar = 1 µm. (**d**,**e**) TEM images of KAIMRC2 cells. Large lipid droplets are apparent inside and outside the cell in the image, d. Scale bar = 2 µm. (**f**) The transmitted light image of KAIMRC2 cells before the transformation. Elongated and fibroblastlike cells are visible. Scale bar = 400 µm. (**g**,**h**) Cuboidal cells after transformation. Scale bar = 400µm. (**i**) Cell size shrinks to 4–5 um, and cells started to make small round colonies (black arrowheads), a typical cancer cell characteristic. Multinucleated giant cells are also visible (white arrowheads), suggesting another subpopulation of cells, probably CSCs. Scale bar = 100 µm. (**j**–**o**) Transmitted light time-series images of adhesion of KAIMRC2 cells on a glass surface. White and black arrows show the formation and circular movement of lipid droplets during cells to the glass surface adhesion process. To the best of our knowledge, this is a very novel mechanism of cell adhesion that has not been shown before. A video of this process is available in the Appendix A.

**Figure 2 cells-10-01303-f002:**
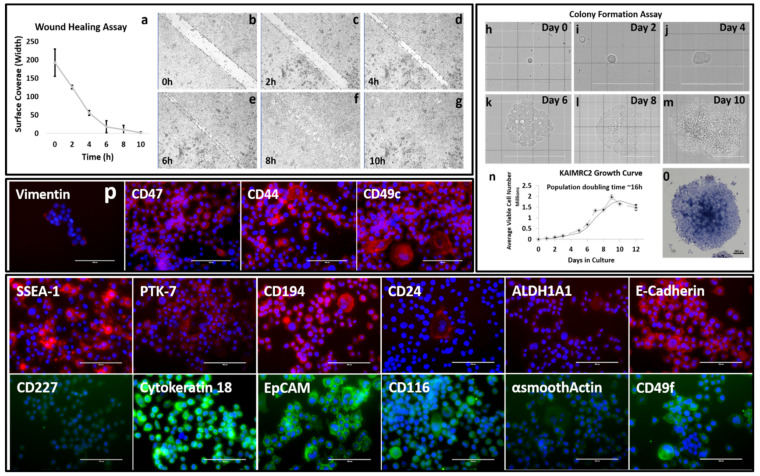
Growth characterization of the KAIMRC2 Cell line. (**a**–**g**) Wound healing assay of the KAIMRC2 cell line. Live-cell time-lapse images of the wound healing assay were acquired. A confluent layer of the cells was scratched (the area between the two parallel dotted lines), and transmitted light time-lapse imaging was performed for 24 h. Images show cells after 0 h, 2 h, 4 h, 6 h, 8 h, and 10 h. Scale bar = 200 μm. (**h**–**m**) Colony formation assay of KAIMRC2 Cell Line. Formation of colonies was visualized microscopically, and transmitted light images were acquired three times a week. Scale bar = 200 μm except m = 400 μm. (**o**) Colonies were stained with Hematoxylin and Eosin on day 10, and images were acquired using Molecular Devices EVOS FL Auto system. Scale bar = 200 μm (**n**) The average growth curves of the cell line are shown. Cells were counted in triplicate for 24 days. The doubling time of KAIMRC2 cell line was approximately 16 h. (**p**) Immunocytochemistry images of a panel of markers to perform a comprehensive proofing of the KAIMRC2 cell line. Scale bars = 100 μm.

**Figure 3 cells-10-01303-f003:**
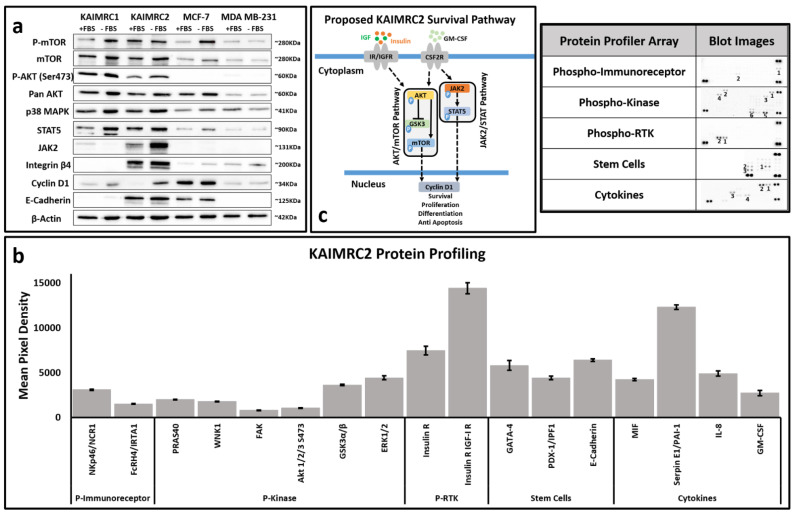
(**a**,**b**) Protein profiling of KAMRC2 cell line. (**a**) Western blot analysis of comparison of several breast carcinoma cell lines revealed that KAIMRC2 has constitutively active *AKT* and *mTOR*. Strongly positive Integrin β4 and E-cadherin are indicative of cancer stemlike subtype. (**b**) Several protein profiler arrays showed noticeable insulin receptor expression, *GATA-4*, *PDX-1*, *E-cadherin*, and *Serpin E1*. Interestingly, high expression of *Cyclin D1* was found in the starvation condition in KAIMRC2 cells. (**c**) Our proposed model of the KAIMRC2 cell line survival pathway.

**Figure 4 cells-10-01303-f004:**
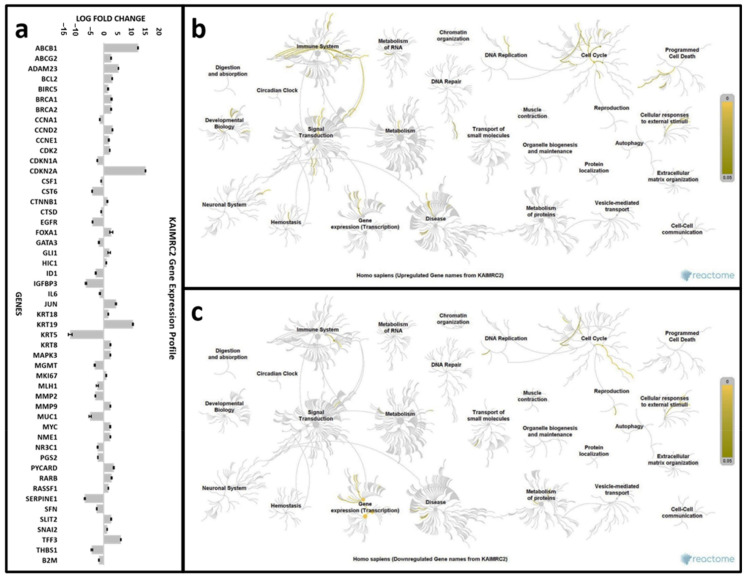
Gene Expression Analysis of breast cancer KAIMRC2 cell line. (**a**) Relative up- or downregulation of key genes of KAIMRC2 cells is presented. Each column represents a single gene and represents data from duplicates. (**b**,**c**) Segregation of the identified genes into up- and downregulated genes. The web-based freely available pathway analysis tool, Reactome Pathway database (reactome.org), was used to identify the pathways affected by these genes.

**Figure 5 cells-10-01303-f005:**
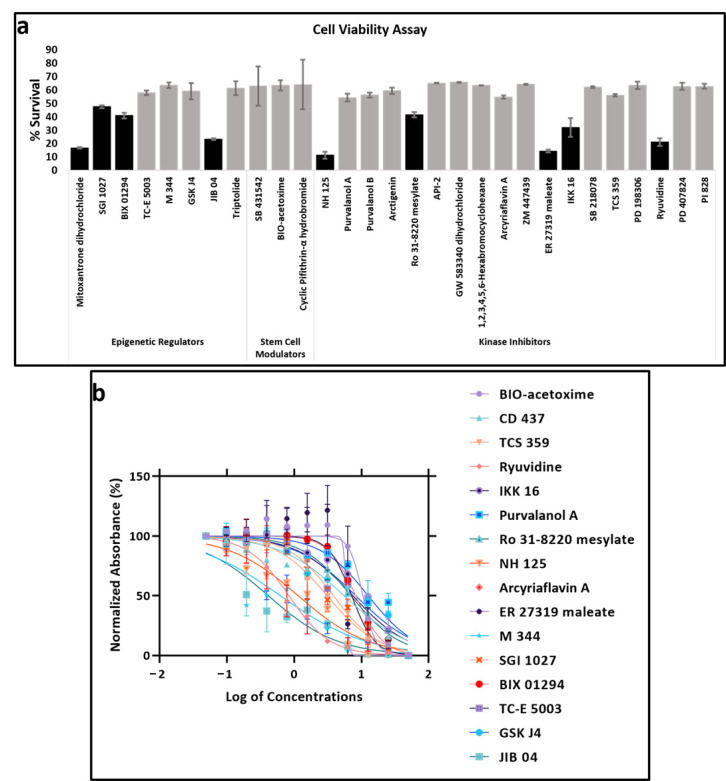
Treatment of KAIMRC2 cell line with kinase inhibitors, stem cell modulators, and epigenetic regulators. Cell viability was assessed by MTT assay. (**a**) Initially, compound panels were tested at 10 µM concentration. Cytotoxic compounds with a % cell survival of less than 50% were selected (Black bars). (**b**) IC_50_ calculation of the selected compounds.

**Figure 6 cells-10-01303-f006:**
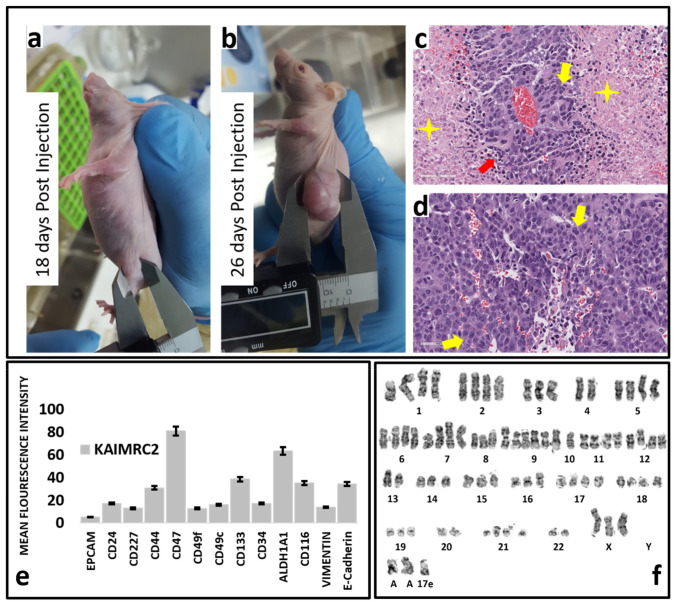
Determination of tumorigenicity potential of the KAIMRC2 cell line. (**a**) Image of nude mice after 18 days of injection. A noticeable tumor lump is visible. (**b**) Image of the same mice on the 26th day. A big lump of the tumor is visible. The mouse was sacrificed on the same day, and the tumor was isolated and stained. (**c**,**d**) H&E staining of a histological section of the isolated tumor. Cells exhibit all the hallmarks of malignancy, i.e., nuclear pleomorphism, hyperchromasia, clumped nuclear chromatin, and mitoses (yellow arrows). Focal tumor necrosis is also present (Yellow 4-point star). Extensive tumor coagulative necrosis and the concentration of viable cancer cells can be seen around the blood vessels, i.e., Peritheliomatous necrosis (red arrows), a hallmark of the high-grade tumor. Scale bar = 50 µm. (**e**) Flow cytometric analysis of selected breast cancer and CSCs markers expressed on the KAIMRC2 cell line. (**f**) Representative karyograph of KAIMRC2 cell line. It revealed the presence of an abnormal, complex, and composite karyotype including chromosome range of 58–79; hexasomy of chromosome 9, tetrasomy of chromosomes 1, 2, 5, 6, 7, 11,12, 17, 18 and 21; trisomy of chromosomes 3, 8, 14, 15, 16 and 19. Genomic instability and the existence of heterogeneity in the KAIMRC2 cell line is represented by these chromosomal abnormalities.

**Table 1 cells-10-01303-t001:** Immunoprofiling of the KAIMRC2 cell line. A strong panel of fibroblasts, epithelial, and stem cell biomarkers was used to characterize the cells by flow cytometry. Strongly positive CD47 and CD49c hints toward this cell line’s epithelial lineage, whereas strongly positive CD44, E-cadherin and ALDH-1A1 indicate CSCs potential of the KAIMRC2 cell line.

Description	Markers	Cell Lines				
		MCF-7	MDA-MB231	MCF10A	KAIMRC1	KAIMRC2
Epithelial, Fibroblasts and Tumor Cell Lines Marker	CD47	***	***	***	***	***
Monocytes, Macrophages and Fibroblasts Marker	CD116	*	-	**	-	**
Monocytes and Granulocytes Marker	SSEA-1	**	-	-	-	*
Surface Marker of Some Breast Cancers	HER-2	-	-	-	-	-
Epithelial Marker	CD49c	**	***	***	-	***
Breast and Colon Epithelial Marker	CD227	**	***	***	-	**
Messenchymal and Fibroblast Cells Makrer	Vimentin	-	-	-	-	**
Marker for Breast Cancer Stem Cells	CD24	***	-	***	-	**
Tumor Epithelial Cells	EpCAM-EBA1	**	-	**	-	**
Epithelial Cells and Cancer Stem Cells Marker	CD44	**	***	***	**	***
Stem Cells and Epithelial Adherent Junctions Marker	E-cadherin	**	-	***	***	***
Cancer Stem Cells and Breast Cancer Cells Marker	ALDH-1A1	***	***	***	**	***

* = Weakly Positive, ** = Moderately Positive, *** = Strongly Positive.

**Table 2 cells-10-01303-t002:** IC_50_ values of selected cytostatic stem cell modulators, kinase inhibitors, and epigenetic regulators on the KAIMRC2 cell line. All these compounds showed IC_50_ values lower than 10 µM. TCS 359, Ryuvidine (a derivative of benzothiazoledione), NH125, M344, SGI 1027, a Quinoline-based DNA Methyltransferase Inhibitor and JIB04 showed IC_50_ values even lower than 5 µM.

Compounds	Target	Biological Description	IC_50_	IC_50_ Range	R^2^
BIO-acetoxime	Glycogen Synthase Kinase 3	Selective GSK-3α/β inhibitor	10.52	9.057 to 11.96	0.9240
CD 437	Retinoic Acid Receptors	Enhances reprogramming efficiency of MEFs	5.540	4.327 to 7.052	0.9528
TCS 359	FLT3	Potent inhibitor of FLT3 receptor tyrosine kinase	3.092	2.602 to 3.664	0.9629
Ryuvidine	cdk	Cyclin-dependent kinase (cdk) 4 inhibitor	0.8339	0.6558 to 1.084	0.9243
IKK 16	IKK	Selective inhibitor of IκB kinase (IKK). Inhibits TNF-α-stimulated IκB degradation and expression of adhesion molecules E-selectin, ICAM and VCAM	7.176	6.024 to 8.406	0.9293
Purvalanol A	cdk	Cyclin-dependent kinase inhibitor	12.61	10.28 to 15.45	0.9176
Ro 31-8220 mesylate	Broad Spectrum Inhibitor	Protein kinase C inhibitor, with activity at other protein. Also inhibits voltage-dependent Na+ channels in the micromolar range.	7.025	5.955 to 8.298	0.9539
NH 125	CaM Kinase III	Histidine protein kinase and eukaryotic elongation factor 2 (eEF-2) kinase (CaMK III) inhibitor. Blocks G1/S cell cycle progression	1.147	0.9333 to 1.409	0.9520
ER 27319 maleate	Syk	Selective inhibitor of Syk kinase	~5.947	Very wide	0.908
M 344	pan-HDAC	Histone deacetylase inhibitor	0.6241	0.3621 to 1.040	0.8225
SGI 1027	DNMT1	DNA methyltransferase inhibitor	3.843	3.279 to 4.513	0.9669
BIX 01294	G9a/GLP	G9a-like protein and G9a histone lysine methyltransferase inhibitor	8.220	7.711 to 8.766	0.9900
TC-E 5003	PRMT1	Selective PRMT1 arginine methyltransferase inhibitor	7.446	6.006 to 9.228	0.9341
GSK J4	JMJD3/UTX	Histone demethylase JMJD3/UTX inhibitor	8.234	6.407 to 10.61	0.9156
JIB 04	pan-JMJD	Pan Jumonji histone demethylase inhibitor	0.4134	0.264 to 0.654	0.8575

**Table 3 cells-10-01303-t003:** Experimental and measurement details of the mice model used to determine the KAIMRC2 cell line’s tumorigenicity potential.

Mice	Size in Length and Width (mm) after 4 Weeks	Size in Length and Width (mm) after 8 Weeks	Tumor Weight (g) after 8 Weeks	Body Weight (g) after 8 Weeks
1	5.5–5.6	6.7–6.3	1.7	24
2	9–7	13–11	1.75	26
3	7–8	9.4–10.3	2.5	30

## Data Availability

All data generated or analyzed during this study are included in this published article (and its Appendix A). The KAIMRC2 cell line will be available upon request to any researcher worldwide through King Abdullah International Medical Research Center (KAIMRC). The rules and regulations of the institution will be applicable.

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
