# Peer review of "Isolation and Establishment of a Highly Proliferative, Cancer Stem Cell-Like, and Naturally Immortalized Triple-Negative Breast Cancer Cell Line, KAIMRC2"

_cells, 2021, doi:10.3390/cells10061303_

Round 1

Reviewer 1 Report

This study describes molecular, biochemical and morphological features of a new triple-negative breast cancer cell (TNBC) line, named KAIMRC2. This cell line expresses key breast epithelial and cancer stem cells (CSCs) markers, including CD47, CD133, CD49f, CD44, and ALDH-1A1. The results of this study also demonstrated a comprehensive gene and protein expression analyses, in which it was revealed these cells overexpress ABC transporter and AKT-PI3Kinase as well as JAK/STAT signaling pathways. The karyotype analysis revealed a large number of chromosomal anomalies, including hexasomy of chromosome 9, tetrasomy of chromosomes 1, 2, 5, 6, 7, 11,12, 17, 18 and 211 21; trisomy of chromosomes 3, 8, 14, 15, 16 and 19. Changes on chromosome 8q, 11p, 11q, 13q, and 17q are very common in breast cancers. Western blot analysis of comparison with MDA-MB-231 and MCF-7 breast carcinoma cell lines revealed that KAIMRC2 has over-phosphorylated STAT5, JAK2, AKT and mTOR proteins. KAIMRC2 cell line shares many molecular markers (mesenchymal, epithelial, stem and fibroblast differentiation marker) present in MCF10A, which is a non-tumorigenic epithelial cell line. The study also shows the results of colony formation in soft agar and the tumorigenicity assay in nude mice. Overall the methods and assays are appropriated and well conduced and the text is well-written.

Critiques:

At the end of the discussion section the authors conclude that KAIMRC2 cell line display cancer stem cell-like features and metastatic potential. However, there is no figure or table showing lung metastases formation in vivo which could prove metastatic potential of cells after xenotransplantation. There are no studies showing the abilities of stem cells to exclude vital dyes based on the expression of ATP binding cassette transporter. Therefore, the authors should extend their studies showing for example, the results of assay to identify KAIMRC2 breast cancer lung colonization or liver/brain/bone seeding and distal metastasis using an orthotopic injection and intravenous (IV) cell injections.  Additionally, the author should perform flow cytometer studies to characterize the CD44high/CD24low expression and functional enrichment analyses using the ALDEFLUOR assay to characterize and isolate stem-like cell population. These studies are likely to provide specific insights into the tumorigenic mechanism of breast cancer cells isolated from a Saudi breast cancer patient suffering from invasive ductal carcinoma.

Author Response

Response: We are thankful for the suggestions made by the Reviewer. Thank you for pointing out the very important lung metastases formation xenotransplantation in-vivo experiment that is lacking in the manuscript. This experiment is lengthy and it was not initially planned in the original project proposal so we did not collect organ samples from the sacrificed animal. This is indeed a very good suggestion and we will include this data in our next manuscript in which we are planning further in-vivo experiments targeting the tumor with the candidate compounds identified in this work.

We have already performed a flow cytometry study to characterize and isolate the CD44+/CD24low KAIMRC2 stem cell-like population. We have added Supplementary figure 4 showing the R&D systems MagCellect™ CD44+/CD24- based human breast cancer isolation of KAIMRC2 cells (Lines 741-743).

Unfortunately, further functional studies were not in the scope of this work and they will be included in other manuscripts. Moreover, we want to point out that we made our conclusions of stemness and metastatic potential of KAIMRC2 based on the colony formation assay, and overexpression of key markers, CD44, CD49c, CD116, ALDH1A1, EpCAM, SSEA-1, and Oct4. We are already performing further studies to substantiate our conclusions and we will publish the results in our next manuscript.

Once again thank you for all the very useful comments.

Reviewer 2 Report

In this study a new cell line from a Saudi Arabian female breast cancer patient was characterized. The primary tumor was an invasive ductal carcinoma which was triple-negative. The cell line KAIMRC2 had stem cell-like characteristics and it was concluded to provide a platform in breast cancer research especially from patients with different ethnic backgrounds.

Comments

  1. It is confusing to the reader that the Figures do not appear in chronological order. Further, Figures 1, 2, 3 and 4 appear two times each. Please rearrange the Figures throughout the manuscript.

  1. Page 4, Line 159. Include (Figures 2a-g).

  1. Page 17, subheading 3.10. Gene expression analysis. Please discuss the fact that the normal breast MCF-10A cells, used as reference cells, had a different ethnic background than the KAIMRC2 cells.

  1. Page 18, Lines 573-574. You state that: “High expression of CD47, CD49c, CD44 and ALDH1A1 confirmed the breast cancer origin of the KAIMRC2 cell line”. Please specify in which way these markers are breast specific. This is not known to this reviewer.

Author Response

We are very grateful to the reviewer for taking the time to read through our manuscript and we thank you for pointing out the mistakes. We have made the necessary corrections to the manuscript.

Comments

  1. It is confusing to the reader that the Figures do not appear in chronological order. Further, Figures 1, 2, 3, and 4 appear two times each. Please rearrange the Figures throughout the manuscript.

 Response: Thank you for the suggestion. We have rearranged the figures and also included the supplementary figures.

  1. Page 4, Line 159. Include (Figures 2a-g).

 Response: It has been corrected in Line 184

  1. Page 17, subheading 3.10. Gene expression analysis. Please discuss the fact that the normal breast MCF-10A cells, used as reference cells, had a different ethnic background than the KAIMRC2 cells.

 Response: We have added a sentence mentioning the ethnic background of the normal breast MCF10A cells in Lines 814-815

  1. Page 18, Lines 573-574. You state that: “High expression of CD47, CD49c, CD44, and ALDH1A1 confirmed the breast cancer origin of the KAIMRC2 cell line”. Please specify in which way these markers are breast-specific. This is not known to this reviewer.

Response: The authors have done an extensive literature search to find out a group of potential markers for Breast origin and we ended up with this list. Although these markers are not only expressing on breast origin cells but they are usually highly expressed on breast origin cells. Below are a few of the publications highlighting the use of these markers in defining the origin of a cell line.

  1. Yuan J, Shi X, Chen C, et al. High expression of CD47 in triple-negative breast cancer is associated with epithelial-mesenchymal transition and poor prognosis. Oncol Lett. 2019;18(3):3249-3255. doi:10.3892/ol.2019.10618
  2. Kong Y, Lyu N, Wu J, et al. Breast cancer stem cell markers CD44 and ALDH1A1 in serum: distribution and prognostic value in patients with primary breast cancer. J Cancer. 2018;9(20):3728-3735. Published 2018 Sep 8. doi:10.7150/jca.28032
  3. Ginestier C, Hur MH, Charafe-Jauffret E, et al. ALDH1 is a marker of normal and malignant human mammary stem cells and a predictor of poor clinical outcome. Cell Stem Cell. 2007;1(5):555-567. doi:10.1016/j.stem.2007.08.014

We are very thankful to the reviewer for the useful comments that helped us to improve our manuscript.